# OpenReview forum: "TAMAS: Benchmarking Adversarial Risks in Multi-Agent LLM Systems"
_ICLR.cc/2026/Conference — ICLR 2026 Conference Withdrawn Submission_

### Official Review · Reviewer_nmYQ · 2025-11-01

**Soundness:** 2
**Presentation:** 3
**Contribution:** 2
**Rating:** 4
**Confidence:** 2

**Summary:**

The paper introduces TAMAS, the first benchmark for evaluating the safety and robustness of multi-agent LLM systems. It argues that existing benchmarks focus only on single agents, failing to capture unique risks from multi-agent collaboration. TAMAS tests systems against six attack types (including prompt injections, impersonation, and multi-agent-specific attacks like colluding or contradicting agents) across five high-stakes domains. The key finding is that multi-agent systems are "highly vulnerable" to these attacks, revealing "new, systemic risks" not seen in single-agent setups.

**Strengths:**

1. The paper introduces TAMAS, the first benchmark to systematically evaluate the safety of multi-agent LLM systems. Its key innovation is defining and testing "multi-agent-specific risks" (like Byzantine, Colluding, and Contradicting agents) , which "have no analog in single-agent setups".

2. The work is methodologically rigorous. The TAMAS benchmark is comprehensive, spanning 300 adversarial instances across five domains and six attack types. The evaluation is thorough, testing 10 LLM backbones across three agent configurations and two frameworks. The introduction of the Effective Robustness Score (ERS) provides a thoughtful metric for the safety-utility trade-off.

3. The paper is well-written and clearly motivated, identifying that multi-agent safety is "largely under-explored". The attack framework is clearly explained with text and an excellent overview figure (Figure 1). The results are presented accessibly in tables and figures, with "Illustrative Cases" in the appendix making the risks concrete.

**Weaknesses:**

The core weakness is the paper's focus on demonstrating failure without providing actionable steps for mitigation or a deep root cause analysis.

* Missing Defenses: The paper does not test the effectiveness of simple, common defenses, like providing agents with explicit refusal instructions (safety guardrails) in their prompts, which limits its practical use.

* Shallow Analysis: It needs a deeper root cause analysis to distinguish whether failures are due to: Model-Level Compliance (LLM ignoring safety training) and Orchestration Failure (the multi-agent framework logic breaking down).

* Scalability Issue: The reliance on human evaluation for labeling the outcomes is slow and unscalable. Reporting Inter-Rater Reliability (IRR) scores and validating against an LLM-as-a-Judge method are needed for community adoption.

**Questions:**

Please refer to the weaknesses part above.

---

> ### Author Response · Authors · 2025-11-20
> **Response to Reviewer nmYQ**
>
> We thank the reviewer for their detailed feedback and are glad that they found our work methodologically rigorous, clearly motivated and well-written. We address all questions and weaknesses below and incorporate the suggested clarifications in the revised manuscript.
>
> **W1: Missing Defenses**
>
> We appreciate the reviewer's suggestion to include defensive baselines.
>
> We will incorporate several lightweight yet commonly deployed defenses and evaluate their effectiveness on the most vulnerable configuration (RoundRobin).
>
> Direct prompt injection: (a) explicit bracketing of user queries using \<USER\>…\</USER\> tags, and (b) semantic paraphrasing of queries to extract the original user query.
>
> Impersonation: paraphrasing the incoming message to remove any authoritative identity cues.
>
> Indirect prompt injection: (a) tagging tool responses with \<TOOL OUTPUT\>…</TOOL OUTPUT> tags, and (b) sandwich defense that appends an integrity-preserving statement after tool outputs.
>
> For interaction-based attacks (contradicting, colluding, and byzantine agents), we will adopt a monitoring-based defense, where system traces are checked by a monitor LLM at frequent intervals to decide whether to continue or terminate execution.
>
> We are currently running these experiments and will provide the full results and analysis during the discussion phase. These additions will offer a meaningful defensive baseline, enabling a clearer assessment of the benchmark’s practical difficulty. While this set is not exhaustive, we note this in the limitations section (Section 9) and use these defenses as practical baselines for the current evaluation.
>
>
> **W2: Shallow Analysis**
>
> In our study, we find that failures in multi-agent systems arise from a combination of model-level limitations and framework-level orchestration failures.
>
> Model-level failures:
>
> Several attack types reveal that the underlying backbone LLM does not reliably apply its safety constraints in multi-agent settings.
> DPI: The backbone LLM fails to distinguish between the user query and the injected instruction, even when it is explicitly irrelevant and/or malicious.
>
> Impersonation: The harmful nature of the user query is ignored in most cases and executed.
>
> Colluding Agents: The model follows malicious agent descriptions unquestioningly, and executes malicious actions without refusals.
>
> Framework-level failures:
>
> At the same time, multi-agent frameworks contribute significantly to the system’s vulnerability:
>
> DPI and IPI: Frameworks pass messages or tool outputs between agents without any validation, allowing unsafe instructions to propagate unchecked.
>
> IPI specifically: Tool outputs are trusted by default. Frameworks implement no guardrails to verify whether a tool response is safe or adversarial, leading the model to misinterpret harmful tool output as legitimate instruction.
>
> Byzantine and contradicting agent attacks: Malicious or disruptive agents continue derailing the conversation until message limits are reached, illustrating that the framework lacks early-termination mechanisms or anomaly detection to halt harmful trajectories.
>
> Both these failure modes reinforce each other, thus amplifying the risks. Unsafe content propagates because frameworks lack checks, and harmful actions are executed because the model fails to refuse them. We have added this additional analysis in the revised version of the manuscript.
>
> **W3: Scalability Issue: The reliance on human evaluation for labeling the outcomes is slow and unscalable.....**
>
> We acknowledge the reviewer’s concern regarding scalability. Human evaluation is commonly used in the community to validate LLM-as-a-judge, and in our work we rely on human annotations only for a targeted subset of outputs to verify the reliability of the judge rather than to label the full dataset. We manually annotated all logs. Each log was independently labeled by two annotators, and any disagreements were resolved by a third annotator who served as the adjudicator. We already report the agreement between human evaluators and the LLM-as-judge in Appendix C.1. In the revised version we additionally include Cohen’s kappa value (0.77) to quantify inter-rater agreement of the human annotators. We believe these additions will address the scalability concern and strengthen the robustness of our evaluation.
>
> **Summary of changes in the revised version:**
>
> **[Section 7]**: Additional analysis to understand the safety failures.
>
> **[Appendix D.2]**: Additional analysis of colluding agents attack.
>
> **[Appendix C.1]**: Additional details about human agreement of LLM-as-judge along with Cohen’s Kappa for inter-rater agreement between human annotators.
>
> Thank you again for your thoughtful comments and suggestions. We hope we’ve addressed all your concerns. We are happy to clarify any additional questions you have and engage in a follow-up discussion.

---

### Official Review · Reviewer_okgF · 2025-11-04

**Soundness:** 2
**Presentation:** 3
**Contribution:** 3
**Rating:** 6
**Confidence:** 4

**Summary:**

This paper introduces the TAMAS benchmark to evaluate the safety of multi-agent system against adversarial attacks. The benchmark tests systems across 5 scenarios with 300 adversarial instances across six attack types. By evaluating ten different LLMs across three agent interaction architectures, the authors demonstrate that current multi-agent systems are highly vulnerable to these attacks, introducing a new metric called the Effective Robustness Score (ERS) to measure the trade-off between safety and task performance.

**Strengths:**

1. Originality and Significance: to the best of my knowledge, this is  the first benchmark to evaluate the safety and robustness of multi-agent LLM systems, especially for >= 3 agents. Also, some attacks specified for MAS, such as Byzantine, Colluding, and Contradicting are also tested.  The topic is also an important topic that the community would be interested in, as it addresses under-explored and systemic risks of MAS.

2. The benchmark provides extensive evaluation, spanning five domains, 300 adversarial instances, 100 harmless tasks and three distinct agent interaction configurations. The analysis as well as the proposed metric provides a nuanced understanding of how architectural choices impact system safety, and can benefit follow up studies.

**Weaknesses:**

Weakness 1: Limited Scope of Adversarial Goals (Disruption vs. Misuse)

A limitation of the benchmark is its focus on attacks that disrupt a given task (e.g., Byzantine, Contradicting agents) or manipulate the immediate output (e.g., prompt injection), rather than testing for more severe, exploitative misuse. The safety community is increasingly concerned with threat actors instrumentalizing systems for inherently harmful, multi-step goals. The current benchmark does not appear to evaluate scenarios such as 1) using a multi-agent system (e.g., with a code agent) to collaboratively generate malware or find exploits. 2) attack a system to post or modifying harmful content on social media. 3) exploiting agent-to-tool interactions to exfiltrate private data. While derailing a task is a valid security concern, these "malicious use" scenarios represent a higher-stakes, more complex class of threats.

Weakness 2: Missing of Defense methods

The paper's central finding is the high vulnerability of multi-agent systems, but this conclusion is drawn in the absence of any standard defensive measures. Many of the tested attacks, particularly prompt injections, are well-known vulnerabilities, and it is common practice to deploy LLM systems with system-level safety filters, input/output sanitization, or "guardrail" models (such as Prompt-Guard). By evaluating the systems in an undefended "vacuum," the paper's results, while striking, may be overstated. The lack of a defensive baseline makes it difficult to assess the benchmark's true, practical difficulty.

Overall, I like the paper and did not find any serious weakness. However, evaluating a benchmark paper is more difficult and I may adjust my review after reading other reviewers comments.

**Questions:**

1. In Equation [4-7], what are the optimized parts to maximize the goal. Are they $\delta_i$? If yes, how to optimize them?

2. When testing the Qwen3 model, is the thinking mode on or off?

2. Would like to see the performance on Claude if resources permits. Claude series should be the safest LLMs for both non-agent and agent use.

---

> ### Author Response · Authors · 2025-11-20
> **Response to Reviewer okgF**
>
> We thank the reviewer for their detailed feedback. We are encouraged that they view our work as important and valuable for the community and that it can help future studies. We address all questions and weaknesses below and incorporate the suggested clarifications in the revised manuscript.
>
> **W1: Limited Scope of Adversarial Goals (Disruption vs. Misuse)**
>
> Our benchmark is designed to evaluate inherent vulnerabilities in multi-agent systems (MAS), specifically how easily the agents in a MAS can be manipulated to deviate from their intended goals. While we do not evaluate full end-to-end malicious use cases (e.g., generating harmful code or executing long-horizon malicious workflows), our attack classes serve as fundamental building blocks for such multi-step malicious behaviors in MAS. These attacks reveal where coordination can be subverted, which is a prerequisite for more complex harmful scenarios.
>
> **W2: Missing of Defense methods**
>
> We appreciate the reviewers’ suggestion to include defensive baselines.
>
> We will incorporate several lightweight yet commonly deployed defenses and evaluate their effectiveness on the most vulnerable configuration (RoundRobin).
>
> Direct prompt injection: (a) explicit bracketing of user queries using \<USER\>…\</USER\> tags, and (b) semantic paraphrasing of queries to extract the original user query.
>
> Impersonation: paraphrasing the incoming message to remove any authoritative identity cues.
>
> Indirect prompt injection: (a) tagging tool responses with \<TOOL OUTPUT\>…\</TOOL OUTPUT\> tags, and (b) sandwich defense that appends an integrity-preserving statement after tool outputs.
>
> For interaction-based attacks (contradicting, colluding, and byzantine agents), we will adopt a monitoring-based defense, where system traces are checked by a monitor LLM at frequent intervals to decide whether to continue or terminate execution.
>
> We are currently running these experiments and will provide the full results and analysis during the discussion phase. These additions will offer a meaningful defensive baseline, enabling a clearer assessment of the benchmark’s practical difficulty. While this set is not exhaustive, we note this in the limitations section (Section 9) and use these defenses as practical baselines for the current evaluation.
>
> **Q1: In Equation [4-7], what are the optimized parts to maximize the goal. Are they $\delta_i$? If yes, how to optimize them?**
>
> Equations (4-7) formalize the structure of the attack but are not used as an optimization objective in our work. While, in principle, parameters such as $\delta_i$ could be optimized to search for stronger adversarial strategies, our benchmark uses human-verified attack instructions for each attack type.
>
> **Q2: When testing the Qwen3 model, is the thinking mode on or off?**
>
> The thinking mode was ON for Qwen3 models.
>
> **Q3: Would like to see the performance on Claude if resources permits. Claude series should be the safest LLMs for both non-agent and agent use.**
>
> We appreciate the suggestion and agree that evaluating Claude models would provide valuable additional evidence, given their strong safety performance. Unfortunately, we were unable to include Claude due to lack of API access and credits.
>
> Thank you again for your thoughtful comments and suggestions. We hope we’ve addressed all your concerns. We are happy to clarify any additional questions you have and engage in a follow-up discussion.

---

### Official Review · Reviewer_EZRz · 2025-11-04

**Soundness:** 3
**Presentation:** 2
**Contribution:** 3
**Rating:** 6
**Confidence:** 3

**Summary:**

The paper focuses on vulnerabilities of multi-agent dynamics and co-ordination, propose a benchmark Threats and Attacks in Multi-Agent Systems (TAMAS), including 5 scenarios, comprising 300 adversarial instances across six attack types and 163 tools, along with
100 harmless tasks. And evaluated 10 models, and 3 MAS frameworks from Autogen and CrewAI. They also propose an Effective robustness score (ERS) for assessing the tradeoff between safety and effectiveness.

**Strengths:**

1, The topic of assessing multi-agent system safety is timely and important.

2, The benchmark includes multiple tasks, multiple constructed prompts, and the corresponding metric. And the evaluation includes multiple agentic structures.

**Weaknesses:**

1, Lack of comparison with other agent safey benchmarks [1,2,3], what's the difference and main contribution compared to these benchmarks?

[1] AgentHarm: A Benchmark for Measuring Harmfulness of LLM Agents.

[2] Agent-SafetyBench: Evaluating the Safety of LLM Agents.

[3] Agent Security Bench (ASB): Formalizing and Benchmarking Attacks and Defenses in LLM-based Agents

2, Lack of attack scenarios. For example, the MAS jailbreak [4], malicious coding behavior [5], and agent conversation behavior [6] are less considered.

[4] Infecting llm agents via generalizable adversarial attack.

[5] Multi-agent systems execute arbitrary malicious code.

[6] Red-teaming llm multi-agent systems via communication attacks.

**Questions:**

1, As described in Appendix C.4, the ERS metric is defined as the harmonic mean of the safety score and performance score, which is a reasonable approach to balance these two objectives. However, the paper would benefit from additional experimental validation of this design choice. Specifically, it would be valuable to see ablation studies or comparative analyses demonstrating that the ERS metric more effectively captures the trade-off between safety and performance compared to alternative formulations that also consider both factors.

2, The current safety evaluation relies on ARIA and LLM-as-a-judge as criteria. While this is a common and practical setup, in certain webpage attack scenarios, there exist more direct and reliable criteria — for instance, whether the agent executes a specific unsafe action (e.g., transferring money to a malicious actor). The paper does not discuss why such criteria were not considered, nor does it provide results using these more concrete safety indicators. Including such analyses could strengthen the validity and interpretability of the safety evaluation.

---

> ### Author Response · Authors · 2025-11-20
> **Response to Reviewer EZRz (1/2)**
>
> We thank the reviewer for their detailed feedback and are glad that they found our work timely and important. We address all questions and weaknesses below and incorporate the suggested clarifications in the revised manuscript.
>
>
> **W1:  Lack of comparison with other agent safety benchmarks, what's the difference and main contribution compared to these benchmarks?**
>
> We have added a table for comparison to prior agent-safety benchmarks in Appendix A.1 in the revised version. Here we clarify the distinction: (i) All prior benchmarks evaluate single agent settings and use the ReAct  framework for simulating the agentic traces. TAMAS evaluates multi-agent setup using real frameworks and across different architectural configurations. (ii) We study attacks studying various components of the system like user prompt, environment and the inter-agent communication. (iii) Several of the attacks we study, such as byzantine agents, colluding agents, and contradicting agents have no meaningful analogue in single-agent LLM evaluations, since they rely on inter-agent dynamics and cooperation failures.
>
> **Table: Comparison of TAMAS with other agent safety benchmarks across various aspects**
>
> | Benchmark                 | Agent Setting | Primary Evaluation Objective                                                | User-prompt Level Attacks| Environment Level Attacks | Agent Level Attacks | Execution Environment        |
> |---------------------------|---------------|-------------------------------------------------------------------------------|-------------------|-------------------|-------------|------------------------------|
> | AgentDojo                 | Single-agent  | Robustness to indirect prompt injection                                      | ✗                 | ✓                 | ✗           | Stateful Environment         |
> | AgentSafetyBench          | Single-agent  | Evaluate safety of LLM-agents across various risk categories                  | ✓                 | ✗                 | ✗           | Simulation through prompting |
> | AgentHarm                 | Single-agent  | Evaluate refusal toward harmful or unsafe user-intended queries               | ✓                 | ✗                 | ✗           | ReAct                        |
> | Agent Security Bench      | Single-agent  | Robustness of LLM-agents across adversarial attacks                           | ✓                 | ✓                 | ✗           | ReAct                        |
> | **TAMAS (ours)**          | Multi-agent   | Robustness of multi-agent LLM systems across adversarial attacks              | ✓                 | ✓                 | ✓           | AutoGen, CrewAI              |
>
>
> **W2:  Lack of attack scenarios. For example, the MAS jailbreak, malicious coding behavior, and agent conversation behavior are less considered.**
>
> Our benchmark is designed to evaluate inherent vulnerabilities in multi-agent systems (MAS), specifically how easily the agents in a MAS can be manipulated to deviate from their intended goals. We consider six attack types targeting various components of the MAS such as user prompt, environment and inter-agent communication. The agent-level attacks such as byzantine, colluding, and contradicting agents probe the core coordination and communication mechanisms of the MAS, similar to the works mentioned. While these papers study full end-to-end malicious scenarios (e.g., infectious jailbreak propagation or execution of malicious code), our benchmark focuses on the fundamental failure modes that enable such complex behaviors to emerge. Although we do not evaluate end-to-end malicious scenarios, our attack classes serve as fundamental building blocks for understanding multi-step malicious behaviors in MAS. As noted in our Limitations section (Section 9), the attacks in our benchmark are not exhaustive, and extending our framework to incorporate more complex, end-to-end malicious use cases presents an important direction for future work.

---

> > ### Author Response · Authors · 2025-11-20
> > **Response to Reviewer EZRz (2/2)**
> >
> > **Q1: As described in Appendix C.4, the ERS metric is defined as the harmonic mean of the safety score and performance score.....**
> >
> > ERS is defined to provide a more fine-grained and balanced view of the safety-utility trade-off than existing alternatives. The closest comparable metric is the net resilient performance (NRP) proposed in AgentSecurityBench, defined as NRP = PNA × (1 − ASR), where PNA represents the performance under no attack (utility) and ASR is the attack success rate. NRP implicitly treats safety as a binary outcome. In contrast, agentic evaluations often produce graded safety behaviors such as delayed refusals or attempted but unsuccessful executions. ERS incorporates these intermediate outcomes through the ARIA scores, which distinguishes four categories: direct refusal, delayed refusal, failed attempt, and successful attack. This avoids collapsing the gradation of scores into a single value and gives a nuanced understanding of safety. Additionally, the harmonic mean used in ERS penalizes imbalance between safety and utility more strongly than multiplicative metrics like NRP. As a result, ERS more faithfully reflects the practical requirement that agents must remain both safe and useful under adversarial conditions.
> >
> >
> > **Q2: The current safety evaluation relies on ARIA and LLM-as-a-judge as criteria....**
> >
> > Our safety evaluation does not rely solely on LLM-as-a-judge. Rather, we adopt a hybrid evaluation strategy that already incorporates direct, reliable and concrete criteria. For all attack types except Byzantine and contradicting agents, we use explicit tool-invocation as the indicator of attack success. This corresponds to a concrete, reliable and more direct indicator of an unsafe action being executed. On the other hand, byzantine and contradicting agents attack could correspond to a number of execution traces, making it difficult to capture the attack success through just tool invocations. For these cases, we use an LLM-as-judge with strict criteria to assess whether the agent meaningfully violated its role or goal. We additionally validate the reliability of the LLM judge using human agreement as described in Appendix C.1. We rely solely on tool invocation to determine attack success for the Colluding Agents attack. Additionally, we improved the prompting strategy for the Contradicting Agents attack to ensure higher reliability, and we report the updated results in the revised manuscript.
> >
> > **Summary of changes in the revised version:**
> >
> > **[Section 6]**: Figure 2. Updated results for colluding and contradicting agents attack with higher F1 score of LLM judge.
> >
> > **[Appendix A.1]**: Table 3. Comparison of TAMAS with other safety benchmarks.
> >
> > **[Appendix C.1]**: Additional details about human agreement of LLM-as-judge along with Cohen’s Kappa for inter-rater agreement between human annotators.
> >
> >
> > Thank you again for your thoughtful comments and suggestions. We hope we’ve addressed all your concerns. We are happy to clarify any additional questions you have and engage in a follow-up discussion.

---

### Official Review · Reviewer_4dV7 · 2025-11-04

**Soundness:** 2
**Presentation:** 2
**Contribution:** 2
**Rating:** 2
**Confidence:** 3

**Summary:**

This paper introduces TAMAS, a benchmark designed to evaluate the adversarial robustness and safety of multi-agent LLM systems. The authors define a threat model encompassing six attack types, including prompt-level injections and attacks described as multi-agent-specific: Byzantine, Colluding, and Contradicting agents. The benchmark dataset spans five domains and is used to evaluate ten different LLM backbones across three distinct agent interaction configurations derived from the Autogen and CrewAI frameworks. The principal finding of the study is that current multi-agent LLM systems exhibit significant vulnerabilities to these adversarial attacks, often failing to refuse explicitly malicious instructions.

**Strengths:**

- The idea is interesting.
- Its studied problem of adversarial vulerabilities in multi-agent systems is interesting.

**Weaknesses:**

- The paper's claims to originality are overstated, it fails to properly clarift what is fundamentally new about their evaluation of adversarial attacks compared against the attacks under the LLM or single agent context.
- The tasks and tools are partly synthetic, so it is unclear how well results transfer to real systems with live APIs and true side effects.
- The quality of the benchmark execution is also questionable; the dataset of 300 adversarial instances seems small for the scope of the claims.
- The safety judging leans on LLMs, yet the paper does not report agreement with humans or how consistent the labels are.
- The combined score is not fully defined and lacks sensitivity checks, and the tables do not report uncertainty or repeat runs.

**Questions:**

- What is the exact formula for your combined score (including weights and any normalization)?
- How does that score change if the mix of benign vs. attack tasks changes, or if you use a different threshold for “attack success”?
- What is the agreement rate between humans and the LLM judge on safety labels, and how did you resolve any conflicts?
- How stable are the results across different random seeds, decoding settings (e.g., temperature), and prompt or role templates, and can you report confidence intervals from repeat runs?
- Can you add defenses suited for multi-agent systems (message checks, permission gates, or consensus filters) and show their effect on both safety and normal task success?
- How do attack success rates change when you vary team size and the communication graph, and when you use another agent framework?

---

> ### Author Response · Authors · 2025-11-20
> **Response to Reviewer 4dV7 (1/3)**
>
> We thank the reviewer for their detailed feedback and are glad that they found our work interesting. We address all questions and weaknesses below and incorporate the suggested clarifications in the revised manuscript.
>
> **W1: The paper's claims to originality are overstated, it fails to properly clarify what is fundamentally new about their evaluation of adversarial attacks compared against the attacks under the LLM or single agent context.**
>
> We clarify that our contribution lies in evaluating safety and robustness in multi-agent LLM systems, which has not been systematically studied in prior work. Existing evaluations primarily focus on single-agent settings or simulate multi-turn reasoning via frameworks such as ReAct. In contrast, TAMAS is the first benchmark to study the robustness of multi-agent systems across various adversarial settings. Attacks considered in our study target various components of the system such as user prompt, environment and inter-agent communication. Several of the attacks we study, such as byzantine agents, colluding agents, and contradicting agents have no meaningful analogue in single-agent LLM evaluations, since they rely on inter-agent dynamics and cooperation failures. We also evaluate across different frameworks (Autogen, CrewAI) and architectural configurations (centralized, decentralized). This allows us to assess how these choices affect the vulnerability of the MAS, an aspect not explored in prior work.
>
> **W2: The tasks and tools are partly synthetic, so it is unclear how well results transfer to real systems with live APIs and true side effects.**
>
> We acknowledge the reviewers concern and clarify that the use of simulated tools is a deliberate design choice to ensure reproducibility, stability, and controlled evaluation. Real-world APIs often undergo updates, temporary outages and rate-limits making the benchmarking non-deterministic and difficult to reproduce. By employing simulated tools, we isolate the agent’s decision-making and actions without any confounding variables. Furthermore, this is a widely adopted practice in various benchmarks [1, 2, 3, 4]. We have added this explanation in Appendix A1. While this is an inherent limitation of current evaluation setups, we view incorporating real APIs while maintaining the stability required for reliable evaluation, as a direction for future work.
>
> **W3: The quality of the benchmark execution is also questionable; the dataset of 300 adversarial instances seems small for the scope of the claims.**
>
> Although the benchmark contains 300 adversarial instances, these examples were carefully constructed to systematically span all attack types evaluated in our study. We focus on providing a controlled set of attacks that expose failure modes of multi-agent LLM systems. Importantly, each instance was human-verified to ensure correctness and adversarial validity. The observed vulnerabilities arise consistently across frameworks (AutoGen, CrewAI) and configurations (centralized, decentralized).
>
> **W4 and Q3: The safety judging leans on LLMs, yet the paper does not report agreement with humans or how consistent the labels are.**
>
> To ensure that our LLM-as-judge evaluation is reliable, we conducted a targeted human annotation study. Specifically, we manually annotated 120 conversation logs sampled across attacks, frameworks, and configurations, and compared these labels against the LLM-as-judge outputs. We manually annotated all logs. Each log was independently labeled by two annotators, and any disagreements were resolved by a third annotator who served as the adjudicator. Cohen's kappa of 0.77 indicates strong inter-annotator agreement, demonstrating that the task is well-defined and that humans are consistent in their judgments. We report both the overall F1 score of the LLM-as-judge and the per-attack values. We obtain an average F1-score of 89.17% across attacks. The attack-wise scores are as follows: DPI: 100%, IPI:100%, Impersonation: 90%, Byzantine agent: 80%, Colluding agents: 90% and Contradicting agents: 75%. All these details are mentioned in Appendix C.1. We rely solely on tool invocation to determine attack success for the Colluding Agents attack. Additionally, we improved the prompting strategy for the Contradicting Agents attack to ensure higher reliability, and we report the updated results in the revised manuscript.

---

> ### Author Response · Authors · 2025-11-20
> **Response to Reviewer 4dV7 (2/3)**
>
> **W5 and Q1: The combined score is not fully defined and lacks sensitivity checks, and the tables do not report uncertainty or repeat runs.**
>
> We clarify that all components of the combined score and supporting metrics are fully defined in Appendix C (we will make this pointer more explicit in the main text).
> We compute a safety score derived from the ARIA scores to evaluate the overall safety of the system. This score combines the ARIA metrics as follows:
>
> $$\text{Safety Score}_\text{attack} = \text{ARIA}_1 + 0.5 \times \text{ARIA}_2 - 0.5 \times \text{ARIA}_3 - \text{ARIA}_4$$
>
> To facilitate comparison and interpretation, we apply min-max normalization over each attack to scale it into the range [0, 100].
>
> Performance under no attack (PNA)  is calculated as:
> $$\text{PNA} = \frac{1}{N} \sum_{i=1}^{N}
> \frac{\text{Number of Tools Correctly Invoked in Task } i}
> {\text{Total Number of Tools Required for Task } i} \times 100$$
>
> To capture the trade-off between safety and utility (PNA), we define the Effective Robustness Score (ERS) as follows:
>
> $$SafetyScore_{overall} = \frac{1}{n} \sum_{i=1}^{n} SafetyScore_{attack_i}$$
>
> $$ERS = \frac{2 \cdot SafetyScore_{overall} \cdot PNA}{SafetyScore_{overall} + PNA}$$
>
>
> Given the scale of experiments, repeat runs of the benchmarks is not feasible. However, to account for statistical variability, we report smoothed bootstrapped estimates of the ARIA metric with 95% confidence intervals across models, attack types, and configurations in Appendix D3. Specifically, we perform smoothed row-wise bootstrapping with 10,000 resamples, where each sample is perturbed using a Dirichlet distribution.
>
> **Q2: How does that score change if the mix of benign vs. attack tasks changes, or if you use a different threshold for “attack success”?**
>
> The combined score is not sensitive to the proportion of benign vs. adversarial tasks. We calculate the safety score and utility score separately, and the effective robustness score (ERS) captures the tradeoff between them by taking a harmonic mean. Changing the mix of benign and attack tasks therefore does not affect the metric, as it does not alter the underlying safety or utility estimates. For all attacks except Contradicting and Byzantine attacks, success is determined by whether the agent invokes an unsafe tool, which is a discrete, unambiguous criterion. For Byzantine and contradicting agents attack, where each datapoint could correspond to a number of execution traces, we use LLM-as-a-judge with clear instructions for success criteria, without relying on any threshold.
>
> **Q4: How stable are the results across different....**
>
> Our dataset spans five domains, each with distinct prompts and role templates, providing natural variation in agent instructions and reducing sensitivity to any single template configuration. To account for statistical variability, we now include bootstrapped estimates of the ARIA metric with 95% confidence intervals across models, attack types, and configurations; these results are reported in Appendix D3. We conduct experiments with temperature as zero following standard practice. Under these settings the results are not affected by random seeds.
>
> **Q5: Can you add defenses suited for multi-agent systems....**
>
> We will incorporate several lightweight yet commonly deployed defenses and evaluate their effectiveness on the most vulnerable configuration (RoundRobin).
>
> Direct prompt injection: (a) explicit bracketing of user queries using \<USER\>…\</USER\> tags, and (b) semantic paraphrasing of queries to extract the original user query.
>
> Impersonation: paraphrasing the incoming message to remove any authoritative identity cues.
>
> Indirect prompt injection: (a) tagging tool responses with \<TOOL OUTPUT\>…\</TOOL OUTPUT\> tags, and (b) sandwich defense that appends an integrity-preserving statement after tool outputs.
>
> For interaction-based attacks (contradicting, colluding, and byzantine agents), we will adopt a monitoring-based defense, where system traces are checked by a monitor LLM at frequent intervals to decide whether to continue or terminate execution.
>
> We are currently running these experiments and will provide the full results and analysis during the discussion phase. These additions will offer a meaningful defensive baseline, enabling a clearer assessment of the benchmark’s practical difficulty. While this set is not exhaustive, we note this in the limitations section (Section 9) and use these defenses as practical baselines for the current evaluation.

---

> > ### Author Response · Authors · 2025-11-20
> > **Response to Reviewer 4dV7 (3/3)**
> >
> > **Q6: How do attack success rates change when you vary team size and the communication graph, and when you use another agent framework?**
> >
> > Our multi-agent setup consists of four agents, and depending on the user query, typically 2-4 agents participate in a task. Since our focus is on understanding how easily an individual agent can be manipulated to deviate from its assigned role, attacks such as Byzantine, colluding, and contradicting agents involve compromising only 1-2 agents. Increasing the team size naturally introduces additional failure points and expands the attack surface; however, our experiments show that even a small number of compromised agents (1-2) is sufficient to reliably cause system-level failure.
> > We assess two widely used multi-agent frameworks, AutoGen and CrewAI. Across attacks, CrewAI exhibits stronger resilience, as reflected in higher ERS scores, indicating that framework-level design choices also influence vulnerability.
> > Regarding communication structure, we evaluate multiple architectural configurations (centralized and decentralized). Centralized configurations tend to be relatively safer, but they also introduce a single point of failure: if the orchestrator agent is compromised, the entire system becomes vulnerable. Different decentralized configurations vary in their robustness. For example, CrewAI’s decentralized setup needs subtasks assigned to corresponding agents, whereas AutoGen’s RoundRobin configuration cycles through agents in a fixed predefined order.
> >
> > **Summary of changes in the revised version:**
> >
> > **[Section 6]**: Figure 2. Updated results for colluding and contradicting agents attack with higher F1 score of LLM judge.
> >
> > **[Appendix A.1]**: Clarification for using simulated tools instead of real-world APIs.
> >
> > **[Appendix D.3]**: Bootstrapped results across attacks, models and configurations with 95% confidence interval.
> >
> > **[Appendix C.1]**: Additional details about human agreement of LLM-as-judge along with Cohen’s Kappa for inter-rater agreement between human annotators.
> >
> > [1] Agent Security Bench (ASB): Formalizing and Benchmarking Attacks and Defenses in LLM-based Agents, ICLR 2025
> >
> > [2] AgentHarm: A Benchmark for Measuring Harmfulness of LLM Agents, ICLR 2025
> >
> > [3] TravelPlanner: A Benchmark for Real-World Planning with Language Agents, ICML 2024
> >
> > [4] API-BLEND: A Comprehensive Corpora for Training and Benchmarking API LLMs, ACL 2024
> >
> >
> > Thank you again for your thoughtful comments and suggestions. We hope we’ve addressed all your concerns. We are happy to clarify any additional questions you have and engage in a follow-up discussion.

---

### Note · Authors · 2026-01-06

I have read and agree with the venue's withdrawal policy on behalf of myself and my co-authors.